# Fibre and extracellular matrix contributions to passive forces in human skeletal muscles: An experimental based constitutive law for numerical modelling of the passive element in the classical Hill-type three element model

Lorenzo Marcucci[1,2]*, Michela Bondì[1], Giulia Randazzo[1], Carlo Reggiani[1,2,3], Arturo N. Natali[2,4], Piero G. Pavan[2,4,5]

1 Department of Biomedical Sciences, University of Padova, Padova, Italy, 2 Centre for Mechanics of Biological Materials, University of Padova, Padova, Italy, 3 Kinesiology Research Center, Garibaldijeva, Koper, Slovenia, 4 Department of Industrial Engineering, University of Padova, Padova, Italy, 5 Fondazione Istituto di Ricerca Pediatrica Città della Speranza, Corso Stati Uniti 4, Padova, Italy

* lorenzo.marcucci@gmail.com

## Abstract

The forces that allow body movement can be divided into active (generated by sarcomeric contractile proteins) and passive (sustained by intra-sarcomeric proteins, fibre cytoskeleton and extracellular matrix (ECM)). These are needed to transmit the active forces to the tendon and the skeleton. However, the relative contribution of the intra- and extra- sarcomeric components in transmitting the passive forces is still under debate. There is limited data in the literature about human muscle and so it is difficult to make predictions using multiscale models, imposing a purely phenomenological description for passive forces. In this paper, we apply a method for the experimental characterization of the passive properties of fibres and ECM to human biopsy and propose their clear separation in a Finite Element Model. Experimental data were collected on human single muscle fibres and bundles, taken from vastus lateralis muscle of elderly subjects. Both were progressively elongated to obtain two stress-strain curves which were fitted to exponential equations. The mechanical properties of the extracellular passive components in a bundle of fibres were deduced by the subtraction of the passive tension observed in single fibres from the passive tension observed in the bundle itself. Our results showed that modulus and tensile load bearing capability of ECM are higher than those of fibres and defined their quantitative characterization that can be used in macroscopic models to study their role in the transmission of forces in physiological and pathophysiological conditions.

**Data Availability Statement:** All relevant data are within the paper and its Supporting Information files.

**Funding:** Lorenzo Marcucci's work is supported by the University of Padova under the MSCA SoE @UniPD programme (Acronym of the project: "Heart Fi-Re", funding: Euros 150.000). The funders had no role in study design, data collection and analysis, decision to publish, or preparation of the manuscript.

**Competing interests:** The authors have declared that no competing interests exist.

# Introduction

The primary function of human skeletal muscle is to create forces of several newtons and movements of several centimetres allowing us to move objects, walk, and breathe. These movements are the result of the interaction of myosin molecules and actin filaments, to convert the chemical energy of a single ATP molecule, of about 85 zJ, into nanometer displacements and pico-newton forces [1]. The amplification of these microscopic active forces, and their passive transmission to the tendons, requires a complex multi-scale structure in muscle architecture. Skeletal muscles are composed primarily of muscle fibres or myofibres, embedded in a composite tissue where blood vessels, nerves and connective cells and fibres coexist. This composite tissue surrounding muscle fibres is generally indicated as extracellular matrix (ECM). Because of the fibres arrangement in whole muscle, active forces generated by myosin motors are passively transmitted to the tendons not only through intrasarcomeric proteins, such as actin and titin, along the longitudinal axis of the fibres but also transversally via integrin and dystrophin complexes to the ECM, which thus play a functional role in determining whole muscle performance [2–4], both in physiological and pathological conditions. For example, in aging or Duchenne muscle dystrophy, it has been shown that the decrease in whole muscle performances is not based on a proportional decrease in the active force generated by single fibres, indicating that a decrease in the efficiency of force transmission is also involved [4–6]. Numerical simulations, by means of Finite Element (FE) models, are a powerful tool for translating the vast experimental data on single fibre behaviour to the macroscopic level, however proper characterization of the mechanical properties of the ECM are also needed to give reliable predictions.

Several studies have been undertaken to characterize the functional role of the ECM in transmitting passive tension at the whole muscle level, although its relative contribution with respect to the intracellular structures, predominantly titin, is still unclear. A seminal study on frog skeletal muscle showed that the passive tension was mainly supported by intracellular component [7], a relatively low influence of the ECM was also reported in mouse skeletal muscle [8] and the higher stiffness of titin respect the ECM was demonstrated in cardiac muscle [9]. On the other side, other studies on mammalian muscles, using decellularized preparations, degradation protocols for titin, or direct comparison between fibres and bundles passive stretches, suggest that the ECM sustain a larger fraction of the total passive stiffness [10–13] and that the passive elastic modulus of fibre bundle is approximately twice that of single fibres [14–17]. A recent work from Brynnel and colleagues, based on a genetically modified mouse model with a stiffer titin isoform, has refreshed the debate showing that titin is the main passive stiffness determinant [18]. A difficulty in determining the relative role in passive force transmission is due to the variability among species [19], muscle types [12] and external factors such as aging [20]. Moreover, very few data is available for human muscles, and usually often focussed on the comparison with severe pathologies [14,21].

The principal aim of this work is to estimate the relative contribution to passive tension of intracellular and extracellular components in human muscles. To this end, we analysed experimental data on passively stretched single fibres after viscous relaxation from human vastus lateralis and proposed an exponential function for their stress-strain relation. We then analysed the passive tension of small bundles of fibres obtained from the same tissue, and we extrapolated the constitutive properties of the ECM subtracting from the total passive tension of the bundle the predicted tension associated to the single fibres. The amount of ECM is commonly supposed to be almost negligible in single fibre preparation, while it has been shown to be present in small bundles samples by scanning electron microscope imaging [22], and was also detectable by light microscopy in our preparation. This approach allowed us to

characterize the ECM stress-strain curve with a second exponential curve. Experimental data on mice showed that the ECM is stiffer in old subject than in young ones [20], therefore, we decided to test muscle form elderly people to maximize the observed force sustained by this component.

The experimental tension-deformation curves that we obtained is helpful to characterize the structural parameters of the passive components in a FE models. Current computational power allows us to build multiscale mathematical models that help in the translational process of deducing whole muscle behaviour starting from experimental data at the cellular level. In particular, FE models based on the Hill-type three-elements model for the single fibre, are useful tools to study whole muscles [23–25], because they can merge the relative simplicity and efficiency of the calculations in single fibre to macroscopic muscle architecture. The Hill-type three elements model describes the fibres as characterized by the contractile element (CE), based on the force-velocity and the length-tension relationships, and the elastic element in series with it (SE), which are both posed in parallel with the third elastic element (PE) [23]. The elements in series, CE and SE, must have the same tension and are related to the active part of the contracting fibres, and can be associated to the acto-myosin complex. In a previous study [26], we characterized the active components of muscle contraction SE and CE and described their behaviour when embedded into a bundle as intermediate step in producing a multiscale model from single fibre to whole muscle. The element in parallel, PE, is related to the passive component of the fibres. Moreover, in FE modelling the fibre element is embedded in a ground matrix usually described as an isotropic elastic continuum [23,24,27]. Its mechanical response is described starting from strain energy function depending on the principal invariants of strain tensor. In some cases, a visco-elastic approach has been adopted to describe its mechanical behaviour potentially accounting for the time-dependent behaviour [28] but, in general, the uncertainty in the experimental data has led to a phenomenological description of this component. Micro-structurally based constitutive models have been proposed by different authors [24,29–31]. These models have the potential to deduce the macro-mechanical properties of muscle tissue starting from the mechanical properties of its constituents, as a way of example the spatial disposition of collagen fibres of ECM with respect to muscle fibres. However, it appears that there is a huge uncertainty about the values of some of these properties and this leads to a wide range of variability of deduced macro-mechanical properties. We propose that our data provide a step forward towards the experimental characterization of the constitutive properties for the passive behaviour in a human muscle bundle, to include both intra- and extra-sarcomeric components (Fig 1). Literature data on FE model, demonstrate that this characterization is not sufficiently supported by experimental data, and the microscopic properties are defined just to match the macroscopic experimental results. As an example, passive tension in the PE is often defined as a function of the active tension in CE [27,28,32,33], while these two parameters are not physiologically related. Moreover, especially in macroscopic models [23–25], PE accounts for both the fibre and ECM passive components in the fibre direction, and the deviatoric part of the stress tensor accounts for some additional components of the ECM. This approach makes it difficult to compare their constitutive properties with the experimental data. In the present study we propose to associate the passive tension components in single fibres to the PE, while the passive stress generated by the ECM will be associated to the ground matrix in which the fibres are embedded.

The ECM component plays an important role in total passive tension generated by bundles of muscle fibres, even at relatively low strains, in accordance with previous data [15,16,34]. Our experimentally derived constitutive properties for both the PE and the ECM can be used in future FE models for macroscopic modelling of muscle mechanics, especially to investigate the biomechanical changes related to ageing or diseases.

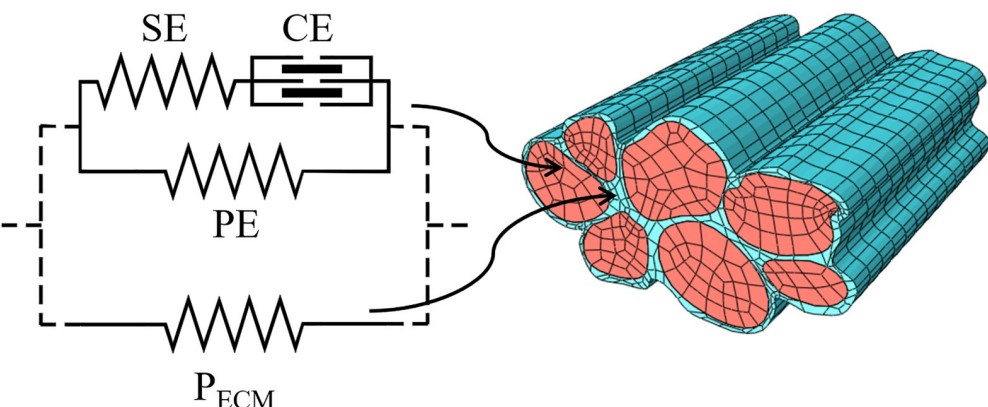

**Fig 1. Scheme of the constitutive model applied to fibres and bundles.** Separated meshes can better describe both intra- (PE, red finite element regions) and extra- (P$_{ECM}$, blue finite element regions) sarcomeric components of the passive forces in muscle bundles through an extension of the classical Hill's three-element model. The experimental characterization of the constitutive laws for PE and P$_{ECM}$ can be made separately, passively stretching single fibres and small bundles. ECM passive properties can be deduced in an almost physiological situation subtracting from the whole passive tension the single fibre component (see text).

## Materials and methods

### Experimental testing of muscle fibres and bundles

Considering the special relevance of ECM for force transmission in muscle aging, and that its stiffness increases with age in mice [20], experimental data were obtained on biopsy samples from *vastus lateralis* muscle of two male healthy donors of 65 and 69 years old in the frame of a project on the impact of aging and activity on skeletal muscles (PANGeA [35]). The study was performed in accordance with the ethical standards of the 1964 Declaration of Helsinki and was approved by the National Ethical Committee of the Slovenian Ministry of Health on April 17, 2012 under the acronym: IR-aging 1200.

Biopsy sampling was done after anaesthesia of the skin, subcutaneous fat tissue, and muscle fascia with 2 ml of lidocaine (2%). A small incision was then made to penetrate skin and fascia, and the tissue sample was harvested with a purpose-built rongeur (Zepf Instruments, Tuttlingen, Germany) [35]. A fragment of the sample was immersed in ice-cold skinning solution and then into storage solution (high potassium, EGTA and glycerol, see for composition [36]) and stored at—20˚C. On the day of the experiment, the biopsy fragment was transferred to a petri dish filled with ice-cold skinning solution, repeatedly washed to remove glycerol and shortly (5 min) bathed in skinning solution containing 1% Triton X-100 to ensure complete membrane permeabilization. Single muscle fibre segments and fibre bundles were manually dissected under a stereomicroscope. Extreme care was taken to avoid overstretching which might disrupt the fibre cytoskeleton or the extracellular matrix. T-shaped aluminium clips (T-clips) were mounted at each end of the fibres or fibre bundles leaving 1–2 mm free and their sides were folded tightly around the fibre or the bundle. Each clip was provided by a small hole so the fibre or bundle could then be transferred to the apparatus (panel a in S1 Fig) and mounted horizontally between two hooks: one linked to the puller (SI, Heidelberg, Germany) to control the length of the specimen and the other to the force transducer (AME-801 Sensor-One, Sausalito, California) (panel b in S1 Fig). The signals from the force and displacement transducers were fed and stored in a personal computer after A/D conversion (interface CED 1401 plus, Cambridge, UK). A total of 11 fibres and 11 bundles were successfully analysed (data reported in S1 File). Bundles were composed of 3 to 6 fibres.

**Table 1. Transversal section areas of bundles.** Transversal section areas of three bundles, estimated from the video camera images assuming a circular cross-sectional area of each fibre and with image analysis on cross cryo-section of the same bundle.

| Bundle # | Video camera measurement of section area ($\mu m^2$) | Cryo-section measurement of section area ($\mu m^2$) | Percentage difference |
|---|---|---|---|
| 1 | 18037 | 20600 | 12.4% |
| 2 | 13018 | 11984 | −8.6% |
| 3 | 19103 | 18384 | −3.9% |

Fibre length was manually adjusted to the shorter unloaded configuration enough to keep the specimen straight. The sarcomere length $L_0$ and the cross-section diameter $d_0$ were measured on images of the central region of the specimen acquired with an optical microscope (Zeiss, Axiovert 35, equipped with a digital camera Optikam B5, OPT, with a magnification of 300x), while the total length of the fibre or bundle between the two T-shaped clips, $L_0$, was measured by direct observation through an optical microscope (Konus Diamond, KONUS at 40x magnification), as shown in S1 Fig. Fibres cross sectional areas (CSAs) were estimated from the measured d0 supposing a circular CSA, while for the bundles we used the sum of the measured single diameters and areas of the individual fibres. This estimate assumes that the layer of the ECM around each fibre is included. In three bundles, we tested the error introduced by the method of cross-sectional area measurement comparing the estimated area with that obtained by direct measurements on images of cryo-cross sections of the bundles, as described below. Average error was about 8% (Table 1). Sarcomere length was measured analysing the images using the software ImageJ [37], considering the total length of an array composed by at least 10 sarcomeres. The measurement was repeated in several places for each fibre and bundle and the average value was then computed. In this way, we obtained the average sarcomere length in each fibre and in each bundle at different length steps.

The fibre, or the bundle, was stepwise passively elongated by about 20% of $L_T$ each step, to a total of more than 200% of $L_T$ (S1 Fig). To analyse the elastic component of the force, separated from the viscoelastic component, we followed a previously proposed protocol [14,20], and, at each step, the passive force, after a viscous recovery, as well as the cross-section and the length of the sarcomeres in the central region were recorded. Differently from previous works, which ended the relaxation after about 120 seconds overestimating the elastic tension of about 10% [14], we waited until the viscous recovery was almost complete, which lasted in some case up to 600 s, (panel c in S1 Fig). At the end of the experiment, the fibre was brought back to $L_0$ and maximally activated. Once maximum stretch was reached, the fibres or bundles were allowed to shorten back to its initial length and transferred to a pre-activating solution for about 30 s, and then to the activating solution to record the maximum isometric active force generated. Force output was recorded at a frequency of 1 kHz by means of a strain gauge force transducer (AME-801; SensorOne, Sausalito, CA), connected with the A-D converter interface and elaborated with Spike® software. During isometric contraction, a quick shortening was applied to reduce active tension to zero, followed by a re-stretch after 5 ms. The curve of active tension redevelopment, interpolated with a single exponential equation, lead to a rate constant $k_{tr}$ which allowed us to distinguish between slow and fast fibres [38].

Differently from other works, all the measures considered in the following were taken by considering sarcomere elongation because the stretch obtained by considering the length of the whole fibre is generally higher than the stretch pertaining to sarcomeres (see S2 Fig), possibly explained by a limited compression and damage in the region of the samples hold by the T-clips and, therefore, by a non-uniform strain field induced on fibres and bundles.

## Cryo-sections

To obtain the cryo-sections, bundles were removed from the set-up at the end of the mechanical experiment and embedded in OCT matrix (Kaltek srl) under a dissection microscope, set at the initial length and then frozen in isopentane cooled in liquid nitrogen. Serial cross sections (8 μm thick) were cut in a cryostat microtome (Slee, London, UK) set at—24 ± 1˚C. Laminin staining was carried out to identify the edge of the fibres and determine the cross-sectional area (CSA) of individual fibres. Muscle sections were incubated for 1 hour at 37˚C with the polyclonal antibody specific for laminin (L9393, Sigma, St. Louis, MO) diluted 1:150 in 5% fetal bovine serum. Laminin staining was revealed with an anti-rabbit Alexa Fluor 488 (goat anti-rabbit IgG Invitrogen) diluted 1:200 in PBS and incubated for 1 hour at 37˚C. To evaluate the amount of ECM, muscle cryostats were stained with picro-syrius red (direct red 80 sigma-Aldrich 365548). Type 1 and type 2A fibres were identified respectively by immunostaining with BA–F8 (diluted 1:100) and SC–71 (diluted 1:10) monoclonal antibodies (University of Iowa, Developmental Studies Hybridoma Bank) and the reactivity was revealed with the secondary antibodies Alexa Fluor goat anti-mouse IgG2b 488 conjugate for type 1 (primary antibody BA–F8) and IgG1 568 conjugate for type 2A (primary antibody SC–71), diluted 1:200, and 6H1 (IgM, supernatant) specific for type 2X, diluted 1:100. Muscle sections were examined in a Leica RD100 fluorescence microscopy equipped with a digital camera. Muscle fibre CSA and whole bundle area were measured on digital photographs using the ImageJ NIH software.

## Constitutive modelling of muscle fibres

Hill's three-element model [39] is assumed to describe the mechanical response of individual muscle fibres. The nominal stress, i.e. force for unit of undeformed cross area, acting on the fibre is written as $P_{fib} = P_p + P_a$, where $P_p$ and $P_a$ are the stress in passive and active condition, respectively. The active stress $P_a$ is associated to the SE-CE element branch and was the focus of a previous work [26]. The passive stress $P_p$ is related to the PE element and is considered in detail in the following. The stretch of PE element is the ratio $\lambda_{fib} = L/L_0$, $L$ being the deformed length of the sarcomere and $L_0$ its initial length. Considering the stiffening behaviour of muscle fibre in response to passive tensile elongation, as shown in our experimental results as well as from the literature [40], we assume the following function to describe the stress term of PE element in elongation:

$$P_p(L) = A\{exp[(L/L_0)^2 - 1] - 1\} \tag{1}$$

depending on the stress-like parameter $A$ and the initial length $L_0$. The previous equation can be used to fit the constitutive model to experimental data, while for its implementation in the framework of a FE procedure is more suitable to adopt the form:

$$P_p(\lambda_{fib}) = \begin{cases} A[exp(\lambda_{fib}^2 - 1) - 1] & \lambda_{fib} \geq 1 \\ 0 & \lambda_{fib} < 1 \end{cases} \tag{2}$$

where it is recognized also that the parameter $A$ is half of the initial tangential elastic modulus, since it can be simply deduced as:

$$\lim_{\lambda_{fib} \to 1^+} \frac{\partial P_p(\lambda_{fib})}{\partial \lambda_{fib}} = 2A \tag{3}$$

Eq (2) reflects the fact that muscle fibres are assumed to have negligible stiffness in compression.

Different numerical formulations for the PE element can be found in the literature. Two of these formulations were considered for comparative purpose with our proposed constitutive model. Tang et al. [27] have assumed that the nominal stress in muscle tissue due to the fibres under passive condition is given by the equation:

$$P_p(\lambda_{fib}) = \begin{cases} 4P_0(\lambda_{fib} - 1)^2 & \lambda_{fib} \geq 1 \\ 0 & \lambda_{fib} < 1 \end{cases} \tag{4}$$

where the stress-like parameter $P_0$ is the maximum isometric stress. Applying the derivative (3), the initial tangential stiffness for this formulation is null. Alternatively, an exponential function has been proposed by Xhang et al. [25] to describe the stress-stretch response of PE element:

$$P_p(\lambda_{fib}) = \begin{cases} 2P_0 aA(\lambda_{fib} - 1)\,exp[a(\lambda_{fib} - 1)^2] & \lambda_{fib} \geq 1 \\ 0 & \lambda_{fib} < 1 \end{cases} \tag{5}$$

where $P_0$ is again the maximum isometric stress and $a$, $A$ is a pair of non-dimensional parameters. In this case the initial tangential elastic modulus is given by $2P_0 aA$. Eqs (4) and (5) will be fitted to experimental data in the following by explicitly introducing, also in this case, the initial length through the definition $\lambda_{fib} = L/L_0$.

## Constitutive modelling of muscle fibre bundles

The passive stress response of muscle bundle is assumed as sum of the contribution of fast fibres, slow fibres and ECM. In the following it is written in terms of the first Piola-Kirchhoff stress tensor:

$$\mathbf{P}_p = (\alpha_f P_{pf} + \alpha_s P_{ps})\mathbf{n} \otimes \mathbf{n}_0 + \alpha_c \mathbf{P}_c \tag{6}$$

In the previous equation the stress $P_{pf}$ and $P_{ps}$ refer to fast and slow fibres, respectively, while $\mathbf{P}_c$ is the tensorial term related to ECM. The scalars $\alpha_f$, $\alpha_s$ and $\alpha_c$ are the volume fractions of fast fibres, slow fibres and ECM in the bundles. The volume fractions respect the relation $\alpha_f + \alpha_s + \alpha_c = 1$. The unit vector $\mathbf{n}_0$ defines the local orientation of muscle fibres in the undeformed configuration, which mapped to its counterpart $\mathbf{n}$ in the deformed configuration through the deformation gradient $\mathbf{F}$. Eq (6) reflects the anisotropic contribution to the stress response given by the passive behaviour of muscle fibre because of their specific spatial orientation. The isotropic stress term of ECM is defined as:

$$\mathbf{P}_c = k_v(J^2 - 1)\mathbf{F}^{-T} + \frac{\beta}{\sqrt{\tilde{I}_1}}\,exp(\sqrt{\tilde{I}_1} - \sqrt{3})J^{-2/3}\left(\mathbf{F} - \frac{1}{3}I_1\mathbf{F}^{-T}\right) \tag{7}$$

Where the first term is related to the volumetric part of deformation and the second term to the volume-preserving one. The scalar $J$ is the Jacobian of the deformation $J = det\mathbf{F}$. Defined the right Cauchy strain tensor $\mathbf{C} = \mathbf{FF}^T$, $I_1 = tr\mathbf{C}$ is the first principal invariant and $\tilde{I}_1 = J^{-2/3}I_1$ its volume-preserving part. The term $k_v$ represents the bulk modulus of ECM and $\beta$ is a constitutive parameter related to the nonlinear stress-strain response of ECM. The iso-volumetric part of the stress term is similar to other formulations proposed in the literature [25,26]. It is assumed that the muscle fibres are perfectly bounded to ECM in a bundle, condition that has been considered as reasonable in healthy conditions [31], as in the case of the sample considered in this work. This assumption can be questionable in limit conditions of high stretches– i.e. beyond the physiologic range of a bundle–or for specific diseases, such as in Duchenne

dystrophy [6,41]. Additionaly, the stress term of ECM could be defined to account for anisotropy also of this component. However, in this work ECM has beeen assumed to be isotropic, since the experimental data considered refer to the same direction, namely that of muscle fibres orientation.

### Fitting procedure

The fitting procedure was based on the minimization of the following error measure, with respect to the parameters $A$ and $L_0$:

$$\Xi = \sqrt{\frac{1}{n_{exp}} \sum_{i=1}^{n_{exp}} \left[ P_{i,exp} - P_{num}(L_{i,exp}, A, L_0) \right]^2} \tag{8}$$

where $n_{exp}$ is the number of experimental data and $P_{i,exp}$, $L_{i,exp}$ are the measured values of nominal stress and sarcomere length, respectively. While $P_{num}$ is the numerical value estimated at the same level of $L_{i,exp}$. The fitting procedure was made by applying a stochastic method implemented in the software Scilab (Scilab Enterprises) with user routines.

We used the Eq (2) to fit experimental data taken from single fibres. A multiple fitting of the same equation was made on experimental data obtained for fast and slow fibres, separately. This allowed to find the best fit for the average response of the groups of fast and slow fibres, as well as to compare the two groups. In addition, we fitted Eqs (4) and (5), taken from the literature, to the same set of data for comparison. Once the parameters $A$ and $L_0$ were obtained for fast and slow fibres, the stress component was deduced for the case of uni-axial stress state along the direction of the bundles, which corresponds to experimental conditions tested. In this case, under the hypothesis of incompressibility, the stress response of bundles is given by:

$$P_p = \alpha_f P_{pf} + \alpha_s P_{ps} + \alpha_c \frac{\beta}{\sqrt{I_1}} \left( \lambda - \frac{1}{\lambda^2} \right) exp\left( \sqrt{\lambda^2 + \frac{2}{\lambda}} - \sqrt{3} \right) \tag{9}$$

with the definition $\lambda = L/L_{0b}$, where $L$ is the current length of the bundle and $L_{0b}$ its stress-free length, i.e. the length for the initial straight configuration of the bundle obtained during the experimental testing before of its effective stretching. It must be noted that in Eq (9) the parameters of the fibres are known from the previous fitting procedures and can be written in general as:

$$P_{pf} = P_{pf}(A_f, L/L_{0f}), \ P_{ps} = P_{ps}(A_s, L/L_{0s}) \tag{10}$$

where the pairs $(A_f, L_{0f})$ and $(A_s, L_{0s})$ are the constitutive parameters obtained for fast and slow fibres, respectively. Furthermore, the volume fractions were determined as average values from cross section images of the bundles, assuming their homogeneity along the length of the bundle. Therefore, in the fitting procedure only the parameter $\beta$ had to be determined by minimizing the error measure:

$$\Xi = \sqrt{\frac{1}{n_{exp}} \sum_{i=1}^{n_{exp}} \left[ P_{i,exp} - P_{num}(L_{i,exp}, \beta) \right]^2} \tag{11}$$

with the same procedure adopted for the muscle fibres.

## Statistical analyses

Statistical analyses for the nonlinear regressions were based on the work of Motulsky and Ransnas [42]. The residual sum of squares was defined for each fitting as $RSS = \Xi^2 \cdot n_{exp}$ while the degrees of freedom was computed as $\nu = n_{exp} - N$, where $N$ is the number of parameters of the equation adopted. To evaluate a possible statistical difference between experimental sets of fast and slow fibres fitted separately by using the same equation, the following values were considered: $RSS_{sep} = RSS_{fast} + RSS_{slow}$, $\nu_{sep} = \nu_{fast} + \nu_{slow}$. Fitting the pooled experimental values of fast and slow fibres with the same equation, the corresponding values $RSS_{pool}$ and $\nu_{pool}$ were obtained and the value for the $F$-distribution

$$F = \frac{RSS_{pool} - RSS_{sep}}{\nu_{pool} - \nu_{sep}} \cdot \frac{\nu_{sep}}{RSS_{sep}} \qquad (12)$$

was compared with the critical value $F$ for $(\nu_{pool} - \nu_{sep}) \times \nu_{sep}$ degrees of freedom at $p$-value of 0.05. A similar procedure was adopted for the analysis of experimental data from muscle bundles.

## Results

### Muscle fibres

Fig 2A shows the comparison of experimental data obtained for fast fibres (n = 4), and slow fibres (n = 7). By a multiple fitting of Eq (1) applied to fast and slow fibres separately, similar pairs of parameters were obtained for the two types of fibres. For fast fibres the pair of parameters was $A = 6.64$ kPa and $L_0 = 2.42$ μm, while for slow fibres the pair $A = 7.01$ kPa and $L_0 = 2.44$ μm was obtained. Statistical analysis on the set of constitutive parameters did not show significant difference between fast and slow fibres (p-value > 0.05), therefore they were pooled together. By the multiple fitting of Eq (1) to pooled fibres it was obtained $A = 6.93$ kPa and $L_0 = 2.44$ μm. A comparison of experimental data of pooled fibres and numerical results obtained with Eq (1) is shown in Fig 2B.

The numerical results obtained with the fitting of Eqs (4) and (5), as two among the stress-elongation functions proposed in the literature [25,27], bring about contradictory results (S3 Fig). Use of Eq (4) imposing physiological values of maximum isometric stress completely fails in the fitting of experimental data of pooled fast and slow fibres; therefore, to improve the fit, it is necessary to leave the maximum isometric stress as free parameter. In this case, the fitting procedure gives $P_0 = 17.20$ kPa and $L_0 = 2.36$ μm. With the use of Eq (5) we can attribute physiological values to the maximum isometric stress, thanks to the presence of parameters $a$ and $A$. Assuming an average value $P_0 = 137.5$ kPa [43], the fitting procedure gives $L_0 = 1.71$ μm, $a = 0.040$ and $A = 0.65$. Notably, the model parameter $A$ has a different physical meaning according to the constitutive formulation it refers to. In addition, it is noted that from Eq (5) the estimated initial length $L_0$ assumes a non-physiological value. In contrast to the previous models [25,27], in our approach the active tension values are not used to define the passive tension behaviour and a single multiplicative, stress like parameter is identified.

### Muscle bundles

The mechanical response of bundles and single fibres to tensile tests is shown in Fig 3. Adopting Eq (1) to fit experimental data for bundles and fibres and comparing the results according to the statistical method described in SI, a strong difference between the two nonlinear regressions (p-value < 0.001) emerged. This means that a further contribution of stiffness must be considered with respect to the stress-strain response of single muscle fibres. We made the

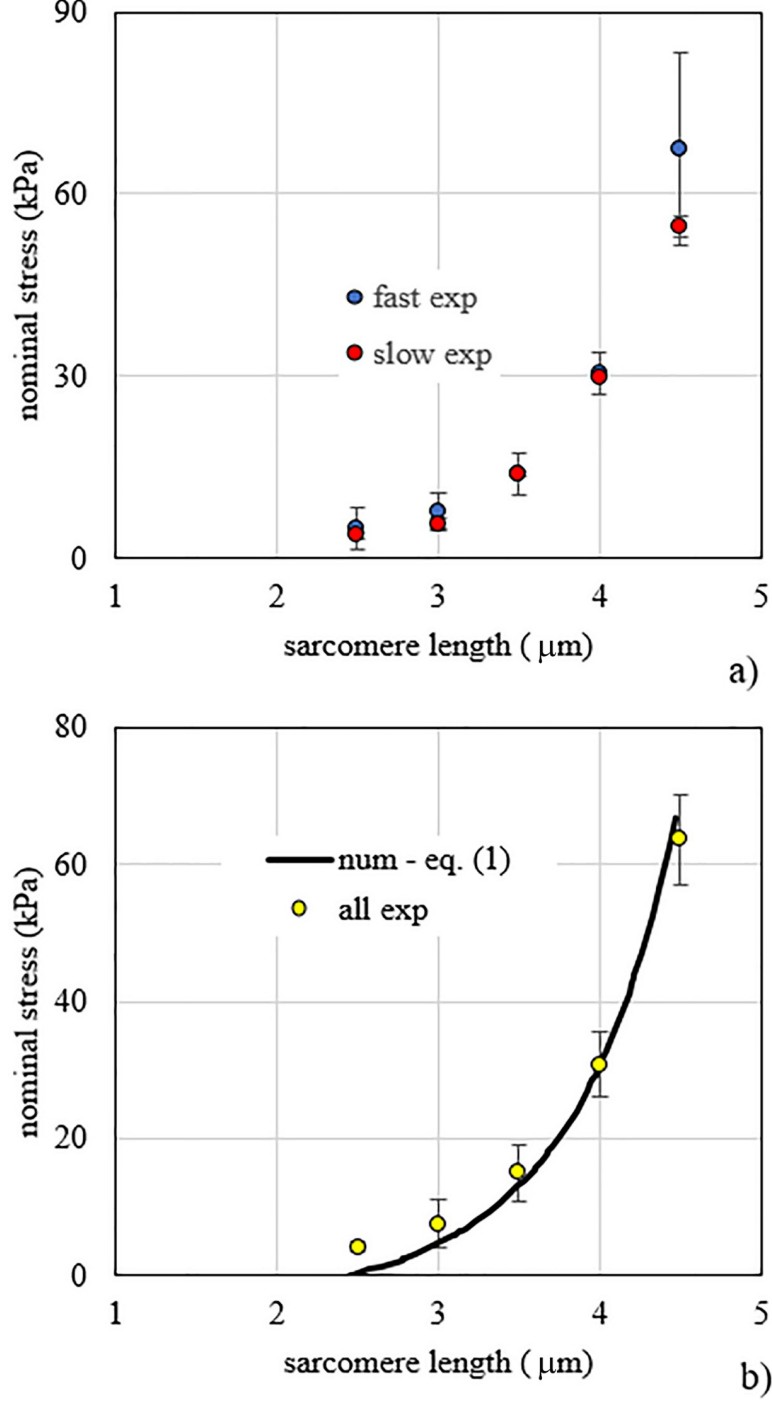

**Fig 2. Fibre stress vs. sarcomere length.** a) Stress vs. sarcomere length experimental data for passive elongation of fast and slow fibres. b) Stress vs. sarcomere length experimental data of pooled fast and slow fibres compared to numerical results obtained by the fitting of Eq (1) proposed in the present work.

hypothesis that this component is due to ECM and we estimated quantitatively its properties by using the Eq (9).

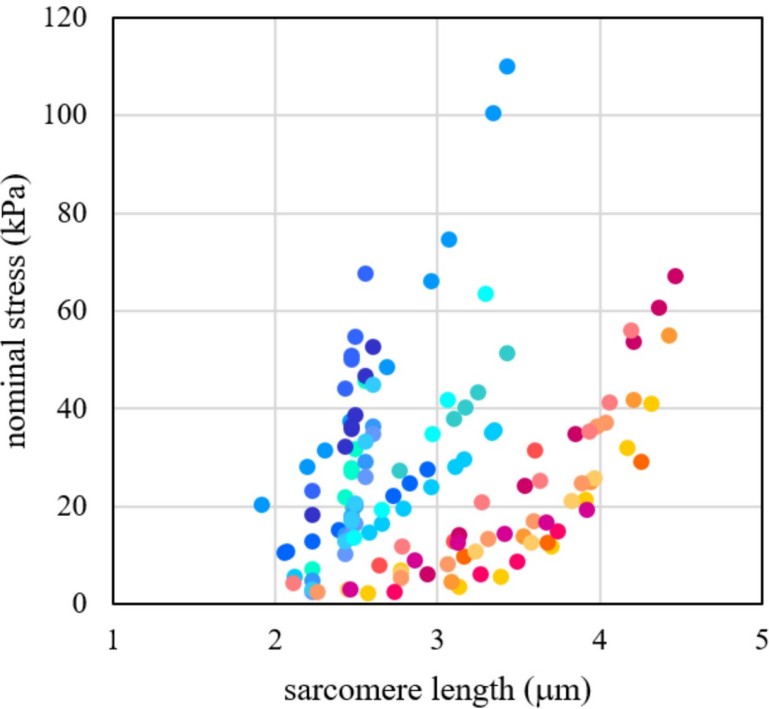

**Fig 3. Bundle stress vs. sarcomere length.** Stress vs. sarcomere length experimental data for passive elongation of single fibres and bundles. Fast and slow fibres are grouped together for comparison. Red tones are for fibres, blue ones for bundles (different colour for each sample).

The fitting of Eq (9) to tensile experimental tests of bundles was made with the set of values $\alpha_f = \alpha_s = 0.425$ and $\alpha_c = 0.15$ for the volume fractions, obtaining $\beta = 168.55$ MPa and $L_{0b} = 1.60$ µm. The choice of the volume fractions was based on data obtained from the analysis of cryo-cross sections in bundles, where the immunohistochemistry data indicated that the ECM corresponded to the 15% of the total CSA. To confirm this value, excluding possible artifacts due to the skinning and dissection procedure, we also estimated the amount of ECM from two cryo-cross sections of intact bundles, i.e. bundles dissected and frozen immediately after the biopsy without undergoing the mechanical experiment (Fig 4). Estimations (19.1% and 21.0%) were compatible, considering the greater fibre cross sectional due to the swelling caused by the skinning procedure [44]. The comparison of numerical results and experimental data are shown in Fig 5, pointing out the contribution of fibres and ECM, separately, and their sum.

To show the different stiffness of fibres and ECM we make use of a tangent stiffness, defined as the slope of nominal stress—sarcomere length curve, which is plotted in Fig 6 versus the sarcomere length values. The nominal stress is calculated for pooled fibres and ECM by considering the force acting on each of them for unit transversal area. The tangent elastic modulus of the whole bundle–defined as slope of the curve of nominal stress versus stretch–is reported in Table 2 at different values of the bundle sarcomere length. The lower cross sectional area of ECM in the bundle with respect to that of fibres is compensated by a higher modulus and the tensile load bearing capability of ECM is always greater than those of fibres, as shown in Table 3 for different values of bundle length.

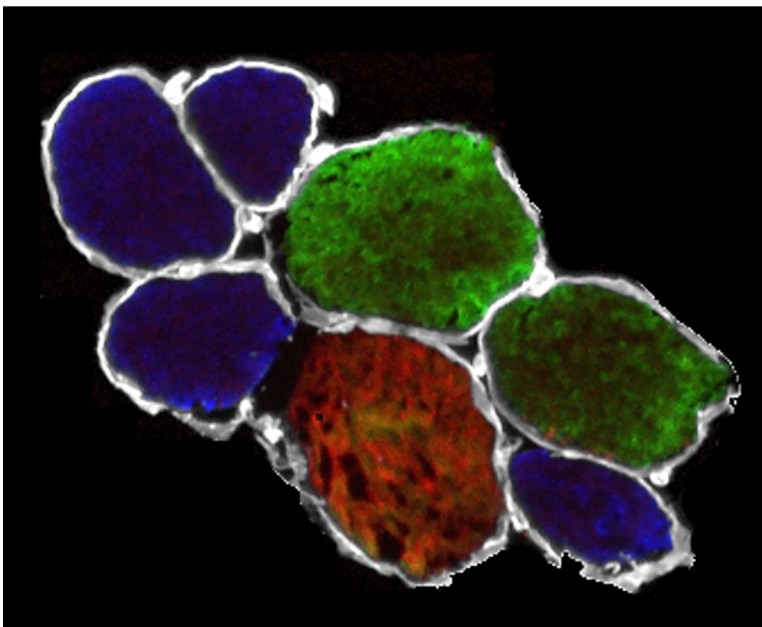

**Fig 4. Cross cryo-section of a bundle.** Cross cryo-section of a bundle not exposed to skinning procedure and mechanical experiment. White colour refers to laminin, blue to slow fibres, green to fast 2A fibres and red to fast 2X fibres.

## Discussion

The relative contribution of Intracellular and extracellular components to the passive stiffness in skeletal muscles is under debate, because experimental data, obtained in various animal species, show a large variability. The variability depends on muscle type [12], species [19] and aging [20], further experimental data on human muscles is needed. Here we analysed the passive tension sustained by the two components in the human vastus lateralis muscle of elderly subjects. Our results indicated that, at each sarcomere length, bundle passive tension was much higher than expected from the sole fibre component, with a high statistical significance. ECM passive tension was derived as the difference between bundle and fibre passive tension at the same sarcomere length. Our results confirmed previous data about the importance of ECM in determining the whole passive structural stiffness within a bundle and showed that for physiological strain ranges ECM contributes at the same order of magnitude of the fibre component to determine the load bearing capacity of a bundle. The ECM stress-strain response was fitted by using a function with exponential behaviour, similarly to muscle fibres.

Our data showed a consistent contribution of the ECM on passive stiffness, similarly to other works in literature [14,16,19,34], based on both animals and humans, but in partial contrast with others as the recent data by Brynnel and colleagues [18], which underline the role of titin. A possible explanation is that we used samples from aged subjects, to maximize the effect of the ECM stiffness, as suggested from experiment on mice [20]. However, our data on bundle tension is consistent with previous values obtained on other human muscles, as shown by comparison in Fig 7, with two representative sets of experimental data from Silldorff et al. [21]. It must be noted that some differences are present between our results and those of Silldorff and co-workers. At the maximum length of bundle elongation, we obtained a tangent elastic modulus of 95.82 kPa. This value is lower than those proposed in that work, i.e. 237.41 ± 59.78 kPa for *supraspinatus* and 193.71 ± 34.96 for *infraspinatus*. The difference can be ascribed to

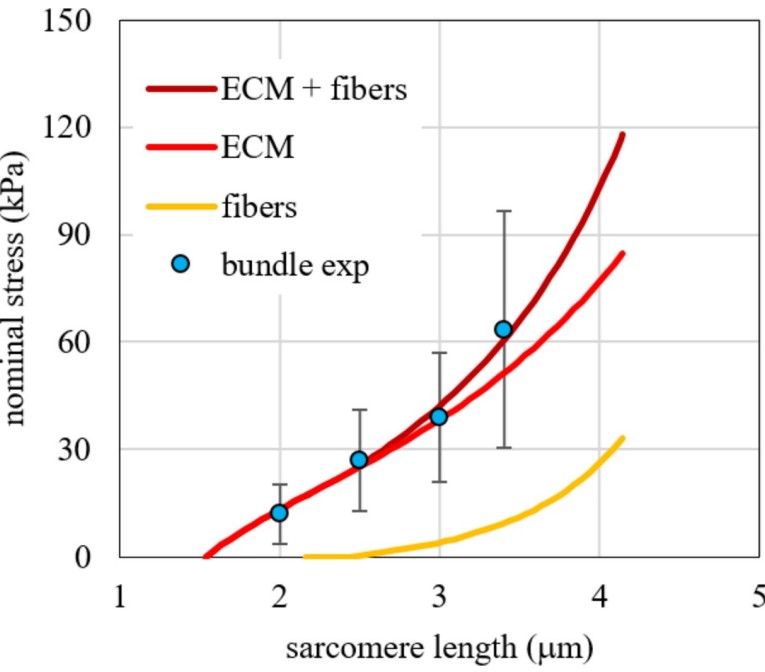

**Fig 5. Comparison between numerical results and experimental data for bundles.** Comparison between numerical results obtained with Eq (9) and experimental data for bundles in terms of force per unit of bundle area vs. stretch. The contribution of all the fibres (yellow continuous line) and ECM (red continuous line) is shown separately and as combination (dark red continuous line).

the following reasons: i) the subjects tested in the two experimental protocols are different and have a relevant difference in age; ii) the muscles tested are different; iii) in their experimental protocol the stress was evaluated at 120 s of relaxation time after each elongation increment, while in our protocol the stress relaxation time is greater–up to 600 s and more–according to the stress drop curve. In particular, the latter can explain a 15% of difference of the tangent elastic modulus, according to the exponential stress drop measured in our test during the stress relaxation phase.

Our data can be used to characterize the embedding matrix in a bundle FE model. In several FE model-based approaches, the constitutive properties of passive elements are defined to fit the macroscopic data for the whole muscle behaviour. However, in some muscle diseases, such as Duchenne Muscle Dystrophy (DMD), or physiological conditions, such as aging, the macroscopic loss of forces is associated to a degradation of both active and passive components of the force at the microscopic level. In all these situations, it is important to have more precise models capable of separating these effects, based on constitutive parameters defined from the experimental data at the fibre/bundle level, and to translate them to the whole muscle. In this perspective, we made a step forward towards the experimental definition of a constitutive function for the passive components in a bundle or a fascicle of fibres which can play as single element of muscle FE models. We proposed a clear separation between components inside the fibres and components outside the fibres (ECM), deducing the former from the single fibre passive behaviour and associating it to the PE in Hill's three-elements model. The latter, due to the ECM in the bundle, was extrapolated from the total passive tension of the bundles and associated to the deviatoric part of the Cauchy tensor, according to the almost-incompressible behaviour attributed to the muscular tissue. The physiological stress-strain curve for the PE element (fibre related components) in a FE model can then be described by Eq (2) with

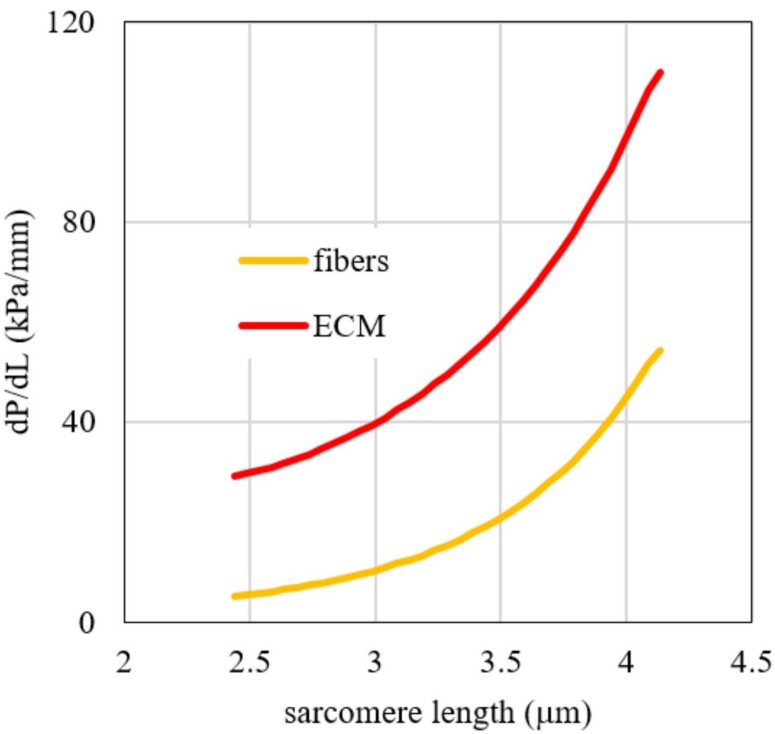

**Fig 6. Tangent stiffness vs. sarcomere length.** Tangent stiffness vs. sarcomere length for fibres (sum of fast and slow) and ECM, as obtained by the fitting procedure of Eq (9) to experimental data of bundles. The tangent stiffness is here defined as the slope of the stress vs. sarcomere length curve of the bundles.

parameters A = 6.93 kPa and $L_0$ = 2.44 μm. The physiological characterization of the deviatoric part of the stress tensor related to ECM component can be described by Eq (7) with parameters $\beta$ = 168.55 MPa and $L_{0b}$ = 1.60 μm. Actually, we observed a more complex behaviour in a broader range of stretching, starting with a quasi-linear region with very low tension, followed by a region of exponential increase up to a failure region, between 3.7 and 5.0 μm, where the tension reaches a plateau or decreases slightly [45]. Recent studies carried out with microendoscopy on human vastus lateralis in vivo showed a working range between 2.8 and 3.2 μm with knee angle changing only from 50° to 110° [46]. This represents only part of the potential whole range of sarcomere length explored in this study, from slack length close to 2.2–2.4 μm to 4.0–4.2 μm which represents the end of the overlap between thick filaments (1.6 μm long in humans as in all vertebrates) and thin filament (1.2–1.3 μm long in humans, see [47]). The

**Table 2. Average tangent longitudinal modulus of bundles obtained from Eq (9), fitted to experimental data, at different values of bundle sarcomere length.**

| Sarcomere length | Tangent elastic modulus |
|---|---|
| (μm) | (kPa) |
| 2.0 | 31.2 |
| 2.5 | 38.4 |
| 3.0 | 51.2 |
| 3.5 | 77.7 |
| 4.0 | 130.7 |
| 4.2 | 155.9 |

**Table 3. Percentage tensile load bearing of ECM and fibres estimated by the fitting of Eq (9) to experimental data.**

| Bundle sarcomere | Force per unit of bundle area | | Tensile load bearing | |
|---|---|---|---|---|
| length | ECM | fibres | ECM | fibres |
| (μm) | (kPa) | (kPa) | (%) | (%) |
| 2.5 | 18.4 | 0.3 | 98.4 | 1.6 |
| 3.0 | 28.3 | 4.0 | 87.6 | 12.4 |
| 3.5 | 40.3 | 11.2 | 78.2 | 21.8 |
| 4.0 | 56.1 | 26.2 | 68.2 | 32.8 |
| 4.2 | 63.8 | 36.4 | 63.7 | 36.3 |

sarcomere lengths explored in this study covered the entire potential range. We determined the stress-strain curves of several slow and fast fibres, taking into account the evidence that differences may exist in passive mechanical properties between slow and fast fibres in relation to the presence of isoforms of titin with different molecular weight [12,45,48,49]. However, no significant difference emerged from the comparison and this allowed us to pool all single fibre curves together. It is worth to recall that the characterization of the constitutive properties of the PE proposed in this work, contrary to previous works, assumed no relationship between passive and active tension in the fibre, coherently with the current knowledge about the molecular mechanisms behind these two components.

Characterization of the ECM properties is a complex task and this work is not fully exhaustive. From experimental tests in animal models, muscle tissue appears to be a non-isotropic material. For example, the passive behaviour of canine diaphragm muscle [50] in biaxial tension shows a greater stiffness transversally to the fibres than along the fibres for nominal strains exceeding 30%, even if this conclusion has not been supported by an adequate number of samples. Moreover, it must be pointed out that those data refer to whole muscle tissue together with myotendinous junction and tendons, not to bundles, and that anisotropy at the two levels of structural hierarchy could be different. However, the few studies reported in the literature about this issue are focused on a strain range that is largely greater than physiological strain, while precise data about a physiological range are not available to the best authors' knowledge. Moreover, there is a major lack of data about human musculoskeletal fibres. For this reason, in this first approach we have adopted an isotropic constitutive model to describe the ECM, considering the mechanical response along the long axis of the fibres to determine the constitutive parameters. It is worth underlining the fact that we made an assumption that fibres are perfectly bound to ECM in a bundle, a reasonably held view in conditions of health [31]. The relevant mechanical properties, such as the shear moduli, are difficult to analyse experimentally. Blemker and co-workers [6,32,33,51] made an extensive numerical analysis of the fibre-ECM interaction at bundle level, especially to determine the ECM shear modulus, but only at theoretical level. Moreover, a variability of stiffness between different muscle types has been observed [21], while here we considered only the *vastus lateralis* muscle.

Despite these limitations, we believe that our results represent an advance in the experimental characterization of the passive properties of individual elements in a muscle FE modelling. Our proposed numerical formulations for the stress-strain relationship for both PE and ECM, are based on human skeletal muscle data, and can be used in future models to reduce the needs of phenomenological approaches. Our proposal to fully separate the intra-fibre and extra-fibre or ECM components in the PE of the three elements Hill's model and strain components acting on the connective component of muscle bundles, makes it possible to treat stress and strain states in the two components separately. This will likely be important when

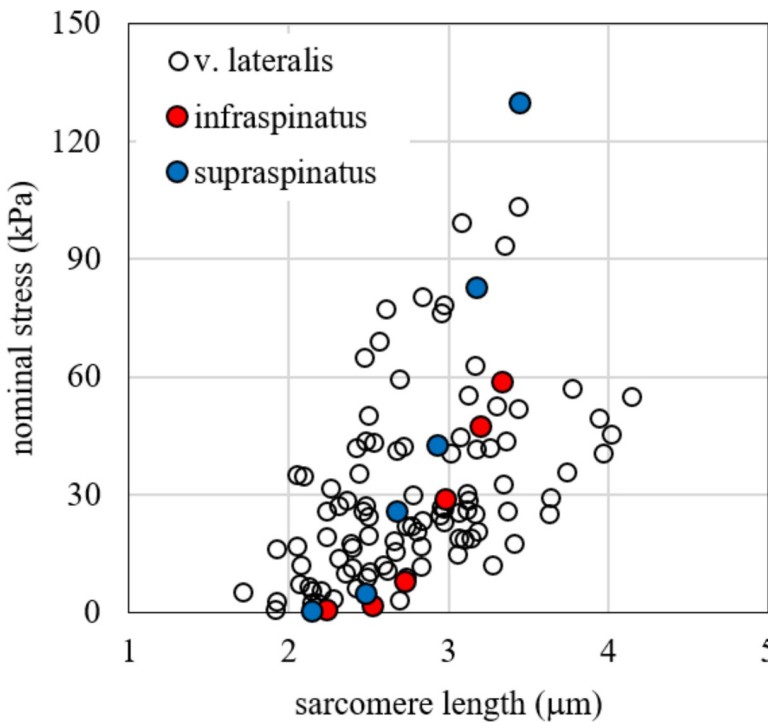

**Fig 7. Stress vs. sarcomere length.** Stress vs. sarcomere length experimental data for passive elongation of bundles tested in the present work compared with experimental data on infraspinatus and supraspinatus found by Silldorff et al. [21].

making predictions concerning pathological degradation of the proteins linking fibre cytoskeleton and ECM. At the single bundle level, a different mesh for the fibres and the ECM may be included. However, in some muscle macroscopic FE models [23,24], the dimensions of the single finite element usually include several bundles of fibres and consequently the contributions of the fibre and ECM components to generate passive tension are not separated. With our approach, a modulation of the two components of the passive force can be included, determining how a change in the ECM proteins and a physiological response of titin, the main determinant of fibre stiffness, modulate the whole muscle behaviour in response to stretch.

## Supporting information

**S1 Fig. Setup for tensile testing of fibres and bundles.** (a). Detail of a fibre bundle mounted with T-clips at its ends (b). Typical stress vs. time experimental data obtained from tests on bundles (c).
(TIF)

**S2 Fig. Stretch measurements on single fibres.** Stretch measurements on single fibres obtained by evaluating length of whole fibre (fibre stretch) and average length of sarcomeres (sarcomere stretch). The straight lines represent equal values for the two measurements. It is noted that the stretch obtained by considering the length of the whole fibre is generally higher than the stretch pertaining to sarcomeres. This could be explained by a limited compression and damage in the region of the samples held by the T-clips and, therefore, by a non-uniform strain field induced on fibres and bundles. Then, all the measures considered in the following were taken by considering the sarcomere stretch. From experimental data, the fibre stretch is

generally higher than sarcomere stretch both for fast fibres (a) and slow fibres (b).
(TIF)

**S3 Fig. Stress vs. sarcomere length experimental data.** Stress vs. sarcomere length experimental data of pooled fast and slow fibres compared to numerical results obtained by the fitting equations proposed in the literature (see Eqs (4) and (5) in the main text of the present work).
(TIF)

**S1 File. Supporting data.** Sarcomere length and tension for each fibre and bundle.
(XLSX)

## Acknowledgments

We thank Dr. Katharine Dyne for checking and improving the English language in the manuscript.

## Author Contributions

**Conceptualization:** Lorenzo Marcucci, Carlo Reggiani, Piero G. Pavan.

**Data curation:** Lorenzo Marcucci, Giulia Randazzo, Carlo Reggiani, Piero G. Pavan.

**Formal analysis:** Lorenzo Marcucci, Michela Bondì, Carlo Reggiani, Piero G. Pavan.

**Investigation:** Lorenzo Marcucci, Carlo Reggiani, Piero G. Pavan.

**Methodology:** Lorenzo Marcucci, Carlo Reggiani, Piero G. Pavan.

**Project administration:** Lorenzo Marcucci, Carlo Reggiani, Piero G. Pavan.

**Software:** Lorenzo Marcucci, Piero G. Pavan.

**Supervision:** Lorenzo Marcucci, Carlo Reggiani, Arturo N. Natali, Piero G. Pavan.

**Validation:** Lorenzo Marcucci, Carlo Reggiani, Piero G. Pavan.

**Writing – original draft:** Lorenzo Marcucci, Carlo Reggiani, Arturo N. Natali, Piero G. Pavan.

**Writing – review & editing:** Lorenzo Marcucci, Carlo Reggiani, Piero G. Pavan.

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
