## [Decision Letter · Decision Letter 0]

12 Aug 2019

PONE-D-19-17455

Fiber and extracellular matrix contributions to passive forces in human skeletal muscles: experimental testing and numerical modelling

PLOS ONE

Dear Dr. Marcucci,

Thank you for submitting your manuscript to PLOS ONE. After careful consideration, we feel that it has merit but does not fully meet PLOS ONE’s publication criteria as it currently stands. Therefore, we invite you to submit a revised version of the manuscript that addresses the points raised during the review process.

We would appreciate receiving your revised manuscript by Sep 26 2019 11:59PM. To enhance the reproducibility of your results, we recommend that if applicable you deposit your laboratory protocols in protocols.io, where a protocol can be assigned its own identifier (DOI) such that it can be cited independently in the future. For instructions see: http://journals.plos.org/plosone/s/submission-guidelines#loc-laboratory-protocols

We look forward to receiving your revised manuscript.

Kind regards,

Jose Manuel Garcia Aznar

Academic Editor

PLOS ONE

a)    Please provide an amended Funding Statement that declares *all* the funding or sources of support received during this specific study (whether external or internal to your organization) as detailed online in our guide for authors at http://journals.plos.org/plosone/s/submit-now.  

b)    Please state what role the funders took in the study.  If any authors received a salary from any of your funders, please state which authors and which funder. If the funders had no role, please state: "The funders had no role in study design, data collection and analysis, decision to publish, or preparation of the manuscript."

Reviewers' comments:

Reviewer's Responses to Questions

**Comments to the Author**

1. Is the manuscript technically sound, and do the data support the conclusions?

Reviewer #1: Partly

Reviewer #2: Yes

2. Has the statistical analysis been performed appropriately and rigorously? 

Reviewer #1: Yes

Reviewer #2: Yes

3. Have the authors made all data underlying the findings in their manuscript fully available?

Reviewer #1: Yes

Reviewer #2: Yes

4. Is the manuscript presented in an intelligible fashion and written in standard English?

Reviewer #1: Yes

Reviewer #2: No

5. Review Comments to the Author

Reviewer #1: Authors present an experimental study on how the skeletal muscle fiber and the extracellular matrix contribute to the passive behaviour of the tissue. The study is conducted on human samples which is very interesting due to the lack of data in the literature nowadays. In spite of that, the way the paper is oriented does not provide enough and proper information for researches in the field of numerical modelling as aimed.

The title includes that numerical modelling has been developed but no results are presented. Also, Finite Element appears in the keywords but only Fig. 1 shows a finite element mesh that is not used later.

Some comments:

Abstract

Line 37: “However, the relative contribution of these two components in transmitting the passive forces is still under debate.” Which components? Active and passive?

Line 45: “…a stress-strain curve which were…” Replace “were” with “was”

Introduction

Line 103-105: “Considering the few data present in literature on human subjects, we decided to test muscle form elder people in this work to increase the amount of force sustained by this component.” This sentence sounds strange to me, please rewrite.

Materials and methods

Experimental testing of muscle fibers and bundles

Line 183: Is there a technical reason why the fiber/bundle was measured stepwise? Are authors aware that this is not a proper response of the tissue for modelling purposes apart from viscoelasticity?

Line 185: Authors should mention here that the fiber recovers its initial length before isometric activation.

Constitutive modelling of muscle fibers

It seems that, according to what is explained in the text, all the passive contribution in the muscle fiber is due to the Parallel Element (PE). Is this correct? What happened with the Elastic Series Element (SE)? The total passive force is the sum of both the PE and SE, are authors neglecting the SE? Where is located the titin in the scheme of the constitutive model in Fig. 1?

Lines 217-218: “Applying the derivative (3), the initial tangential stiffness for this formulation is null.” Something is missing, the initial slope of eq (4) (lambda_fib = 1) is 8*P_0

Line 260: Remove one “definition”

Constitutive modelling of muscle fiber bundles

Eq. (6) The dyadic product should only include the orientation vector in the reference configuration, otherwise is nonsense.

This equation represents an atypical way to define the behaviour of the muscle tissue in contrast to the derivation of the constitutive relationship from a strain energy density function.

Results

Muscle fibers

Lines 289-301: According to my previous comment, it would be surprising that with an extension (passive) test, authors could determine the maximum isometric stress. The conclusion here is that more parameters are needed to fit the passive response of the tissue.

“…in our approach the active tension values are not used to define the passive tension behaviour and a single multiplicative, stress like parameter is identified.” So, why then P_0 is still in eq. (5)?

The overall paragraph is confusing if eq. (4) and (5) are not used in the rest of the work and the results discussed are in the form of supplementary material.

Muscle bundles

Fig.3: The data showed in this form is confusing, please connect the points that belong to the same sample.

On the other hand, sarcomere length is different for each sample for both fibers and bundles, why? Are values obtained in Tab. 2 and Tab. 3 interpolated?

Line 316: Why using the volume fractions? They do not appear in eq. (8) and how was the fitting performed if eq. (8) represents the deformation energy? Is eq. (9) perhaps?

Discussion

Lines 396-398: This last sentence is illegible.

Lines 411: “Here we focused only in the exponential part, considering it the most meaningful for the FE modelling, the main scope of this work” This sentence has to be further explained and proved. Does the muscle physiologically undergoes such levels of deformation (up to 200%)? I disagree that the main scope of the work is FE modelling, the results and conclusions are oriented to compare passive properties of ECM and fibers.

Reviewer #2: What are the main claims of the paper and how significant are they for the discipline?

The paper claims that passive tension generated by the vastus lateralis of aged humans comes primarily from the extracellular matrix, rather than intracellular proteins, as evidenced by studies of passive tension produced by single muscle fibers and bundles of muscle fibers. Additionally, the paper claims that the finite element model presented represents a significant improvement on past finite element models of passively stretched muscle, largely because it more accurately represents the physiological relationships of the parallel, series, and contractile elements typically included in Hill-type models of muscle physiology.

The determinants of passive force in stretched skeletal muscle remain under debate. This study provides one example of a muscle in which the extracellular matrix dominates the passive mechanical properties of the tissue. Such studies can provide insight into the contributions of different components of muscle to passive muscle mechanics and, when viewed in the aggregate with similar studies, will provide insight into variation in the determinants of passive muscle mechanics across vertebrates.

Are the claims properly placed in the context of the previous literature? Have the authors treated the literature fairly?

Yes. A good overview of literature exploring the sources of passive tension in muscle is provided. Evidence is provided for both the extracellular matrix and the intracellular proteins as the primary contributor to passive tension in muscle, and this accurately represents the state of knowledge within the field of muscle physiology. A description of the use of finite element models to study muscle is also included.

Do the data and analyses fully support the claims? If not, what other evidence is required?

Yes. The higher stress generated by fiber bundles relative to single muscle fibers at the same sarcomere length indicate that the extracellular matrix material present in fiber bundles is responsible for most of the passive tension generated.

Do any methods warrant publication of protocols or algorithms as supplementary material? If a protocol is provided, are there any important deviations from it?

No. In its current form, the paper details many important methods of the study in the included supplemental document, but I see no reason why these methods could not be included in the main body of the paper. Moving the methods sections included in the supplemental material into the main body of the manuscript would improve the readability of the paper.

If the paper is unsuitable for pub in its present form, does the study show significant potential that it should be resubmitted?

The paper shows significant potential. The main barriers to publication are stylistic. There are enough errors in grammar and punctuation that they became distracting, and sometimes make the content difficult to understand. There also appear to be some residual edits that were not removed by the authors in the main body of the manuscript. Words that have been struck through with a text editor remain. Use of a professional editing service, such as https://authorservices.springernature.com/language-editing/ could help remedy these issues and improve the readability of the paper.

Additionally, the graphic depicted in figure 1 is not described in sufficient detail and figure 5 contains an unlabeled trace. Finally, as stated above, the methods presented in the supplemental material should be included in the main body of the text to improve readability and allow the manuscript to stand alone.

While these stylistic and organizational concerns reduce the readability of the paper, the question asked by the paper, the experimental methods, and the analytical approach are all strong, and warrant publication.

Are data deposited in appropriate repositories?

It is stated that all data are fully available without restriction, but a repository used is not specified.

Are methods details sufficient for reproduction?

Yes, but only when the supplemental data files are considered. The methods section from the supplemental data file should be merged with the main body of the manuscript so that the paper can stand alone without the supplemental material.

Is the manuscript organized and written clearly enough to be accessible to non-specialists?

Yes. The organization of the paper was logical. Addressing grammatical concerns at the sentence level will improve readability.

Specific comments:

Abstract:

Page 3 Line 37: The abstract refers to the relative contributions of two components in transmitting passive force, but it is not clear which two components are being referenced. Three components are listed immediately before this statement: intra-sarcomeric proteins, fiber cytoskeleton, and ECM.

Introduction:

Page 6 line 111: the contractile element is influenced by a length-tension relationship as well, why is only the force-velocity relationship referenced here?

Page 7 line 144: Could the caption for figure 1 include a description of what the graphic is depicting, how the graphic was generated, and what specifically the arrows are pointing to? What are the grid-like lines overlaying the extracellular matrix depicting?

Page 8 lines 146-150: These sentences seem like a summary of results and potential significance. Would these be better suited to the results or discussion section?

Materials and Methods:

Page 8 line 162: If the epimysium is included, were the samples taken at the muscle perimeter? Do you mean perimysium here?

Page 8 line 163: Could you elaborate on how fibers were dissected? How was fiber damage avoided?

Page 8 line 169: This is the first mention of T-shaped clips. Could you define them for readers that aren’t familiar with this technique?

In general, it seems like a lot of the relevant methods details have been relegated to the supplemental material. Is there a reason that information on the experimental setup, cryo-sectioning and staining approach, and statistical analysis is not included in the main document?

From the methods it is not entirely clear why slow and fast twitch fibers needed to be distinguished. Do these display distinct mechanical properties, and can you provide a citation indicating this?

Results:

Line 327: What is the yellow trace on figure 5? I can assume it is stress from individual fibers, but this is not indicated in the figure legend or caption.

Discussion:

Page 18 line 362 “It likely depends on muscle type, species and aging” can you provide citations for these statements?

Figure 7: Surpaspinatus is misspelled in the figure legend

6. PLOS authors have the option to publish the peer review history of their article (what does this mean?). If published, this will include your full peer review and any attached files.

Reviewer #1: No

Reviewer #2: No

---

## [Author Response · Author response to Decision Letter 0]

20 Sep 2019

We would like to thank you the reviewers for taking the time and effort in reviewing our manuscript. The manuscript has been revised to clarify and address all the reviewers' concerns and comments. The point by point responses are included in the "Response to the reviewers" file.

---

## [Decision Letter · Decision Letter 1]

9 Oct 2019

Fibre and extracellular matrix contributions to passive forces in human skeletal muscles: an experimental based constitutive law for numerical modelling of the passive element in the classical Hill-type three element model

PONE-D-19-17455R1

Dear Dr. Marcucci,

We are pleased to inform you that your manuscript has been judged scientifically suitable for publication and will be formally accepted for publication once it complies with all outstanding technical requirements.

With kind regards,

Jose Manuel Garcia Aznar

Academic Editor

PLOS ONE

Additional Editor Comments (optional):

Reviewers' comments:

Reviewer's Responses to Questions

**Comments to the Author**

1. If the authors have adequately addressed your comments raised in a previous round of review and you feel that this manuscript is now acceptable for publication, you may indicate that here to bypass the “Comments to the Author” section, enter your conflict of interest statement in the “Confidential to Editor” section, and submit your "Accept" recommendation.

Reviewer #1: All comments have been addressed

2. Is the manuscript technically sound, and do the data support the conclusions?

Reviewer #1: Yes

3. Has the statistical analysis been performed appropriately and rigorously? 

Reviewer #1: Yes

4. Have the authors made all data underlying the findings in their manuscript fully available?

Reviewer #1: Yes

5. Is the manuscript presented in an intelligible fashion and written in standard English?

Reviewer #1: Yes

6. Review Comments to the Author

Reviewer #1: All my comments and suggestions have been considered in the new version of the paper and the authors have successfully improved the presentation of their work.

7. PLOS authors have the option to publish the peer review history of their article (what does this mean?). If published, this will include your full peer review and any attached files.

Reviewer #1: No

---

## [Editor Report · Acceptance letter]

15 Oct 2019

PONE-D-19-17455R1 

Fibre and extracellular matrix contributions to passive forces in human skeletal muscles: an experimental based constitutive law for numerical modelling of the passive element in the classical Hill-type three element model 

Dear Dr. Marcucci:

I am pleased to inform you that your manuscript has been deemed suitable for publication in PLOS ONE. Congratulations! Your manuscript is now with our production department. 

With kind regards,

on behalf of

Dr. Jose Manuel Garcia Aznar 

Academic Editor

PLOS ONE